# The Impact of the COVID-19 Pandemic on Physical Activity and Social Isolation among Adults with Physical Disabilities Living in Canada and The Netherlands

Kim Meijer [1], Trynke Hoekstra [1,2], Pim Brandenbarg [2,3], COVID-19 Disability Survey Group [†], ReSpAct 2.0 Group [†] and Femke Hoekstra [2,4,*]

1 Department of Health Sciences and Amsterdam Public Health Research Institute, Vrije Universiteit Amsterdam, 1081 HV Amsterdam, The Netherlands
2 Department of Rehabilitation Medicine, University of Groningen, University Medical Center Groningen, 9700 RB Groningen, The Netherlands
3 Department of Human Movement Sciences, University of Groningen, University Medical Center Groningen, 9700 RB Groningen, The Netherlands
4 School of Health and Exercise Sciences, University of British Columbia, Kelowna, BC V1V 1V7, Canada
* Correspondence: femke.hoekstra@ubc.ca
† Membership of the Group is provided in the Acknowledgments.

**Abstract:** *Background:* The impact of the COVID-19 pandemic among people with physical disabilities might differ between countries due to differences in implemented measures and infection rates. This study aimed to understand the impact of the pandemic on physical activity (PA) and social isolation among adults with physical disabilities in Canada and the Netherlands, and examine associations between PA and social isolation. *Methods:* Secondary data from two studies were used: the Canadian COVID-19 Disability Survey (n = 353) and the Dutch Rehabilitation, Sports and Active lifestyle (ReSpAct) 2.0 study (n = 445). Self-reported PA was measured using IPAQ-SF and Adapted-SQUASH. Social isolation was measured using the PROMIS Social Isolation. Descriptive and regression analyses were performed. *Results:* Canadian participants spent on average 163 min (Median = 0; IQR = 120) on moderate-to-vigorous PA per week and Dutch participants 934 min (Median = 600; IQR = 1125). In Canada, 64% reported to have become less physically active since the pandemic compared to 37% of Dutch participants. In both samples, no clinically relevant associations were found between PA and social isolation. *Conclusions:* The findings emphasize the negative impact of the pandemic on PA and social isolation in adults with physical disabilities in Canada and the Netherlands. Future research is needed to better understand if and how PA can be used to reduce social isolation in people with disabilities. This study illustrates how cross-country collaborations and exchange provide opportunities to inspire and learn from initiatives and programs in other countries and may help to improve PA support among people with disabilities during and after the pandemic.

**Keywords:** physical activity; social isolation; COVID-19 pandemic; physical disability; rehabilitation; mental health

## 1. Introduction

In response to the COVID-19 pandemic, countries all over the world implemented public health and social distancing measures, aiming to slow down the transmission of the virus [1]. For example, in Canada, recreation facilities, national parks, and non-essential businesses were closed for various periods in the first and second year of the pandemic. Citizens were encouraged to maintain a distance of 2 m to others, were asked to avoid social gatherings and crowded places, and were asked to stay at home as much as possible [2]. As many health and social distancing measures were determined by provincial governments, COVID-19 strategies differed within the country [3]. In contrast, the Netherlands implemented measures on a national level. Measures included social distancing,

such as keeping 1.5 m distance to others, closing public places (schools, restaurants and non-essential businesses), and prohibiting large gatherings for various periods in the first and second year of the pandemic [4]. Citizens were asked to stay at home as much as possible [4]. Furthermore, many sports events were canceled [5].

Although these measures were important in limiting the spread of the coronavirus, they have shown to have had negative side effects on physical activity (PA), social isolation, general health, mental health, and lifestyle of people with and without disabilities [6–10]. For example, the closure of many recreation and fitness facilities limited people to engage in PA [11–13]. Results from a rapid systematic review showed that decreased PA during the pandemic was associated with increased levels of depression and anxiety [14]. Similar results were found in another systematic review on relationships between various components of mental health distress (e.g., social isolation, anxiety, stress) and PA in the general population [8]. Furthermore, PA has been suggested as a promising strategy to reduce feelings of social isolation and mental health distress [8]. For example, the Broaden-and-Build Theory of Positive Emotions suggests that individuals' positive emotions and feelings resulting from PA participation may help them to think more creatively and build personal resources, which may then reduce feelings of social isolation [15,16]. Another theory, called the Social Control Theory, suggests that someone's behaviour is partly influenced by their social environment. According to this theory, social control is an important factor in promoting healthy behaviour, such as engaging in PA [17]. In other words, individuals with lower social control as a result of poor social connections (e.g., not being married) tend to engage in less PA compared to individuals with better social control (e.g., married, being socially connected to friends). Taking together, this may suggest a potential negative impact of the pandemic on PA and social isolation. While previous studies have focused on understanding the impact of the pandemic on PA and social isolation [14,18–20], and examined the relationships between these two constructs [21,22], these studies mainly derived from studies conducted in general population or older adults [21,22]. Still, limited studies reported on the impact of the pandemic on PA and social isolation in a diverse group of adults with physical disabilities. Indeed, people with disabilities (PWD) have been considered as an under-research population in PA research [23].

This is concerning, as the pandemic has, in particular, had a negative impact on PA in people with disabilities (PWD) [24]. PWD have been overly affected by the pandemic, as their unique barriers and needs have not always been considered in the response strategies to the COVID-19 outbreak [1,25]. Before the pandemic, studies consistently reported that PWD are less physically active compared to people without disabilities [23,26]. The low levels of PA in PWD can be explained by the many barriers they experience to engage in PA [23]. For example, barriers can exist on community level (e.g., lack of accessible equipment), institutional level (e.g., lack of disability-specific knowledge on PA among healthcare professionals), interpersonal level (e.g., lack of social support), and intrapersonal level (e.g., perceived fatigue or pain) [23]. The pandemic might have worsened already existing barriers as the COVID-19 measures lead to a diminished access to sport and recreation services and limited opportunities to take part in (group-based) PA.

Besides the impact of the pandemic on PA, a scoping review showed that as a result of COVID-19 measures, there are health and social participation differences experienced by PWD compared to people without disabilities during the first wave of the COVID-19 pandemic [27]. For example, PWD experienced limited access to health, education, and essential community services, as well as psychological consequences resulting from disrupted routines and activities [27]. A study in Canada among people working with a disability showed that people with a physical disability reported more health concerns during the pandemic than people without disabilities [28].

While the impact of the COVID-19 pandemic on PA and health in PWD has been described in the literature [9], limited studies have focused on understanding the impact on PA and social isolation in PWD living in different countries. An example of a Canadian cohort study of PWD is the COVID-19 Disability Survey. This study was initiated to

record the experiences, needs and concerns of PWD living in Canada during the pandemic. The survey included questions on PA and social isolation during the pandemic. Another example of a cohort of PWD is the Dutch Rehabilitation, Sports and Active Lifestyle (ReSpAct) cohort study, which started in the year 2012. This cohort included >1700 adults with physical disabilities and/or chronic disease who received support to become and stay physically active during and after rehabilitation, as part of the Rehabilitation, Sports and Exercise (RSE) program [29]. The ReSpAct cohort is known as a relatively active cohort of PWD. During the pandemic, ReSpAct participants were invited to complete a survey on PA and social isolation [30]. We do not know whether ReSpAct participants are able to sustain their high levels of PA during pandemic. While both cohort studies are conducted in different countries with different purposes and historical backgrounds, they have collected similar information on the impact of the pandemic on PA and social isolation in different groups of PWD, making them in particular suitable and interesting for a multi-country study on the impact of the pandemic.

Understanding the impact of the pandemic on PWD living in different countries is of additional value as it provides opportunities to study influence of the pandemic in different contexts and enhances international collaborations and exchange. Furthermore, studying the impact of the pandemic on PWD living in Canada (COVID-19 Disability Survey) and the Netherlands (ReSpAct) is of particular interest, as both countries are high-income countries that play leading roles in expanding our knowledge on rehabilitation, PA and disabilities sports [31,32]. A previous policy comparison study showed that federal governments of both countries provide funding to local governments to promote PA in PWD, but the way they promote PA differs between governments [33]. For example, in Canada national PA promotion programs are organized by provinces and territories via long-term agreements. In the Netherlands, PA promotion in PWD is being done by municipalities via time-limited programs (e.g., PA promotion programs in rehabilitation) [33]. Furthermore, the Netherlands is a small, densely populated country with 17 million people (population density: 512 per km$^2$), while Canada is a large, thinly populated country with more than 33 million people (population density: 3 per km$^2$). Countries also differed in implemented COVID-19 measures and infection rates. On 23 June 2021, the Netherlands counted a total of 1.7 million infections, while Canada, with almost twice the population, counted 1.4 million infections [34]. These differences in PA promotion strategies, infrastructure, and the COVID-19 measures and infection rates might have resulted in differences in perceived impact of the pandemic between people with physical disabilities living in Canada and the Netherlands.

The primary aim of this study was to understand the impact of the COVID-19 pandemic on PA and social isolation among adults with physical disabilities living in Canada and the Netherlands. The secondary aims were to understand the impact of the pandemic on general health, mental health and lifestyle among adults with physical disabilities, and to examine relationships between PA and social isolation. Our hypotheses were that the pandemic has a negative impact on PA, social isolation, general health, mental health and lifestyle in PWD in Canada and the Netherlands; and that the negative impacts are greater in the Canadian sample compared to Dutch sample due to the different historical background of both samples. We also hypothesized that PA and social isolation are significantly associated, in which adults with higher PA levels perceive lower levels of social isolation. This study provides new insights into the impact of the pandemic on this vulnerable group in two different countries. The findings can be used to inform future research and program development to improve PA support to PWD during and after the pandemic.

## 2. Materials and Methods

This secondary data analysis study used a quantitative cross-sectional study design using data from two studies: the COVID-19 Disability Survey in Canada (www.disabilitysurvey.ca, accessed on 7 December 2022) and the ReSpAct (Rehabilitation, Sports and Active Lifestyle) 2.0 study in the Netherlands [30,35]. The Research Ethics Board of the

University of British Columbia gave approval for the COVID-19 Disability Survey (Open Science Framework: https://osf.io/z4gr2, accessed on 7 December 2022). The Central Ethics Committee of the University Medical Center Groningen (UMCG) gave approval for the ReSpAct 2.0 study.

### 2.1. Canadian Study Design and Study Population

The Canadian COVID-19 Disability Survey (www.disabilitysurvey.ca, accessed on 7 December 2022) is an ongoing study consisting of an online 20-min initial survey and four 15-min follow-up surveys. The survey included questions about the experiences, concerns, and needs of people living with a disability during the COVID-19 pandemic in Canada. The COVID-19 Disability Survey was an initiative led by the Abilities Centre in partnership with Canadian Disability Participation Project (CDPP) researchers. Partner organizations from across Canada were involved in the development of the survey. Participants of the COVID-19 Disability Survey were recruited via social media and via the partner organizations.

Adults who identified as having any form of disability, parents of a child with a disability and family members of someone with a disability living in their household were eligible to participate. The term 'disability' refers here to a broad definition of disability, including impairment to see, impairment to hear, impaired ability to walk, impaired flexibility, impaired hand function, impaired memory, chronic pain, learning disability, autism spectrum disorder, intellectual disability, and a psychological, psychiatric or mental health condition. Additional inclusion criteria were being able to either understand written English or French, or American Sign Language (ASL) and being a Canadian resident.

For this study, data from adults with a physical disability who completed the English version of the COVID-19 Disability Survey were used. This included people with an impaired ability to walk, impaired flexibility, impaired hand function and chronic pain. In total, 353 people met the inclusion criteria.

### 2.2. Dutch Study Design and Study Population

ReSpAct is a prospective cohort study. ReSpAct 2.0 is a 6–8 year follow-up study of the ReSpAct 1.0 study [35]. The ReSpAct study was designed to evaluate the RSE program, which is an evidence-based tailored counseling program to promote PA among people with physical disabilities and/or chronic diseases during and after rehabilitation [29]. Participants of the ReSpAct 1.0 were enrolled 3–6 weeks before discharge from rehabilitation. Participants who took part in the ReSpAct 1.0 study were recruited for the ReSpAct 2.0 follow-up study via a newsletter. Further details on recruitment methods are described in Supplementary File S1.

The study population of ReSpAct 2.0 included adults with physical disabilities and/or chronic diseases that participated in the ReSpAct 1.0 study 6–8 years ago. Inclusion criteria of ReSpAct 2.0 were: (1) previous participation in ReSpAct 1.0; (2) being 18 years or older; and (3) having a physical disability and/or chronic disease. Exclusion criteria were: (1) not being able to complete the questionnaires even with help; (2) having withdrawn consent to participate in follow-up research; and (3) being deceased between ReSpAct 1.0 and ReSpAct 2.0.

All participants of ReSpAct 2.0 were included in current study if they were diagnosed with one or more of the following conditions: (1) impairment in musculoskeletal system; (2) amputation; (3) brain impairment; (4) neurological impairment; (5) spinal cord injury; (6) impairment in organs; (7) chronic pain; and (8) other impairments. People with visual or auditive impairments were excluded. People of whom the type of disability/chronic disease was not registered were also excluded. In total, 445 people met the inclusion criteria.

### 2.3. Data Collection in Canada and The Netherlands

Canadian data were collected between 18 December 2020 and 23 August 2021. Dutch data were collected between 29 March 2021 and 26 August 2021. Both studies included

questions about demographics (gender, marital status, age), PA, social isolation, general health, and changes in PA, lifestyle and mental health since the start of the pandemic.

### 2.3.1. Physical Activity

Two validated questionnaires were used to measure self-reported PA levels. In the Canadian dataset, the International Physical Activity Questionnaire Short-Form (IPAQ-SF) was used [36]. The IPAQ-SF is a 7-item survey that assesses moderate and vigorous PA, walking, and sitting time of individuals during the past 7 days. For the Canadian study, the IPAQ-SF was adjusted for people with physical disabilities by including examples of PA specific for people with disabilities, such as wheelchair racing and arm cranking. In the Dutch dataset, the Adapted-SQUASH (Adapted Short Questionnaire to ASsess Health enhancing PA) was used [37]. The original SQUASH is a self-reported recall questionnaire to assess daily PA of adults without a disability based on an average week in the past month [38]. It measures the frequency, duration, and intensity in 4 different settings of PA, namely 'PA during commuting', 'PA at work or at school', 'PA during leisure-time', and 'carrying out household activities'. The Adapted-SQUASH was developed to create a questionnaire that better aligns with the perceived intensity of activities among adults with disabilities by using appropriate metabolic equivalent of task (MET) values [39,40]. The total minutes of moderate and vigorous PA per week was used as the PA outcome measure. For both datasets, three PA outcome measures were reported: the total minutes moderate PA per week, total minutes vigorous PA per week and total minutes on moderate and vigorous PA per week.

In both studies, the impact of the pandemic on PA was measured by asking participants whether or not their PA had changed since the start of the COVID-19 pandemic, using 3 answer options: (1) no change in PA (2) becoming more physically active or (3) becoming less physically active. In Canada, change in PA was asked with 1 general question asking about the change in participation in exercise, sports, and recreational PA. In the Netherlands, this was asked with 4 questions on change in different settings/domains of PA to align with the format of the Adapted-SQUASH (commuting; work/school; leisure time; household activities). For the purpose of this study, we only present the data on change in leisure time PA, as change in leisure time PA is most comparable with the Canadian question on change in participation in exercise, sports, and recreational PA.

### 2.3.2. Social Isolation

In both studies, social isolation was measured using the validated PROMIS Social Isolation 8a questionnaire. The PROMIS Social Isolation assesses perceptions of being avoided, excluded, detached, disconnected from, or unknown by others in the adult population [41]. The questionnaire consists of 8 questions with 5-point answer scale (never; rarely; sometimes; often; and always). Each answer corresponds to a score of 1–5, producing the continuous social isolation raw-score (sum of response scores). A conversion table was used to translate the raw-score into the standardized social isolation T-score. The PROMIS Social Isolation can be used to compare the data of the survey to that of the general population, where a T-score of 50 represents the mean score in a sample of individuals from the general population in the United States (reference population). A higher score indicates greater social isolation [41].

### 2.3.3. General Health, Mental Health and Lifestyle

In Canada, general health was measured using the validated PROMIS Global Health questionnaire [42]. This questionnaire can be used to measure the health of adults and children in the general population and of those living with a chronic condition, and aims to globally reflect someone's assessment of their health. The PROMIS Global Health consists of 10 questions. We used the first item on self-reported general health (the first question). In the Netherlands, general health was measured with one question on their general health, based on the RAND-12 Health Status Inventory, which is an abbreviated version of the

RAND-36 Health Status Inventory (RAND-36) [43,44]. In both studies, the question on self-reported general health used a 5-point answer scale (excellent; very good; good; fair; and poor).

In both studies, the extent to which the pandemic impacted participants' mental health was measured using a 5-point answer scale (not at all; very little; to some extent; to a great extent; and completely). To measure the effect of the pandemic on lifestyle, both studies asked participants whether their eating habits, smoking habits, and alcohol consumption had changed since the start of the pandemic, using a 3-point answer scale (no change; positive change (eating healthier/smoking less/drinking less alcohol); and negative change (eating less healthy/smoking more/drinking more alcohol). These lifestyle factors were chosen as in addition to physical inactivity, poor eating habits, smoking, and excessive alcohol consumption are important risk factors for the development of chronic conditions [26]. The questions on changes in mental health and lifestyle were developed by the Canadian research team and translated into Dutch for the ReSpAct 2.0 survey.

### 2.4. Data Analyses

All analyses were performed in IBM SPSS Statistics for Windows, Version 25.0. IBM Corp, Armonk, NY, USA. Descriptive analyses were used to describe demographic factors and the impact of the pandemic and associated measures on PA, social isolation, general health, mental health and lifestyle.

### 2.4.1. Physical Activity

For both the Canadian and Dutch samples, the results on PA levels were presented as 'moderate PA in minutes per week', 'vigorous PA in minutes per week', and 'total minutes of moderate and vigorous PA per week', giving the mean, standard deviation, minimum, and maximum. Since this variable was not normally distributed, the median, 25th percentile, and 75th percentile were presented as well. For both studies, participants that state to have spent 6720 min or more on PA per week (or 16 h per day) were excluded [37,38].

For the Canadian sample, change in PA was presented by describing the percentage of either no change in PA, becoming more physically active or becoming less physically active regarding participation in exercise, sports, and other recreational PA. In the Netherlands, change in PA was presented by describing the percentage of either no change in PA, becoming more physically active or becoming less physically active during leisure time.

### 2.4.2. Social Isolation

Results of the social isolation raw-score were presented by giving the mean, standard deviation, minimum and maximum of the social isolation raw-score. Since this variable was not normally distributed in the Netherlands, the median, 25th percentile and 75th percentile were also presented. The social isolation T-score was divided in groups of a T-score below 40, between 40 and 50, between 50 and 60, and 60 and higher, to compare this score to that of the reference population.

### 2.4.3. General Health, Mental Health, and Lifestyle Changes

Results were presented by giving percentages on how many people rated their general health as 'excellent, 'very good', 'good', 'fair', or 'poor'. Results for the extent to which COVID-19 negatively impacted one's mental health were also presented by giving percentages on how many people indicated the negative impact on mental health as either 'not at all', 'very little', 'to some extent', 'to a great extent' or 'completely'. Change in lifestyle results were presented by giving percentages on how many people experienced no change, a positive change, or a negative change since the start of the pandemic for each lifestyle aspect.

2.4.4. Associations between Physical Activity and Social Isolation

The associations between PA and social isolation were analyzed using linear regression analyses, conducted separately for both samples. Since the score of the Dutch sample was not normally distributed, bootstrapped analyses were conducted (bias corrected and accelerated; 1000 samples) for that sample. The results of the (bootstrapped) analyses are presented using a table showing the regression coefficient (B), *p*-value, and 95% confidence interval of both the crude and adjusted analysis (confounders included age, gender, marital status). See Supplementary File S2 for a detailed substantiation of these confounders.

**3. Results**

*3.1. Characteristics of the Study Population*

Table 1 presents participants' descriptive information of the Canadian sample (n = 353) and Dutch sample (n = 445). Of the Dutch participants, 49.7% identified as man, compared to 34.1% of the Canadian participants. In both studies, most of the participants were aged between 51 and 70 years old. In the Dutch study, the majority was married or living as if married (69.6%), while most of the Canadian participants was divorced, separated, never married, or single (59.2%).

**Table 1.** Participants' demographics per country (short version).

| Descriptive Variables | Canada (n = 353) | The Netherlands (n = 445) |
|---|---|---|
| **Gender [n (%)]** | Missing n = 7 | Missing n = 99 |
| Man | n = 101 (29.2%) | n = 172 (49.7%) |
| Woman | n = 228 (65.9%) | n = 170 (49.1%) |
| Not listed/prefer not to answer | n = 17 (4.9%) | n = 4 (1.2%) |
| **Age in categories [n (%)]** | Missing n = 5 | Missing n = 0 |
| 18–30 (young adults) | n = 40 (11.5%) | n = 14 (3.1%) |
| 31–50 (middle-aged adults) | n = 141 (40.5%) | n = 119 (26.7%) |
| 51–70 (senior adults) | n = 146 (42.0%) | n = 266 (59.8%) |
| ≥71 (elderly persons) | n = 21 (6.0%) | n = 46 (10.3%) |
| **Marital status [n (%)]** | Missing n = 0 | Missing n = 100 |
| Married or living as if married | n = 122 (34.6%) | n = 240 (69.6%) |
| Widowed | n = 17 (4.8%) | n = 8 (2.3%) |
| Divorced, separated, never married, or single | n = 209 (59.2%) | n = 86 (24.9%) |
| Other | n = 5 (1.4%) | n = 11 (3.2%) |
| **Type of disability * [n (%)]** | | |
| **(Possible to have more than 1 disability)** | Missing n = 0 | |
| Impaired ability to walk | n = 266 (75.4%) | - |
| Impaired flexibility | n = 161 (45.6%) | - |
| Impaired ability to use hands | n = 125 (35.4%) | - |
| Chronic or long-term pain | n = 240 (68.0%) | - |
| Other | n = 53 (15.0%) | - |
| **Diagnosis/disability main category ** [n (%)]** | | Missing n = 0 |
| Impairment musculoskeletal system | - | n = 76 (17.1%) |
| Amputation | - | n = 17 (3.8%) |
| Brain | - | n = 120 (27.0%) |
| Neurology | - | n = 71 (16.0%) |
| Spinal cord injury | - | n = 10 (2.2%) |
| Organs | - | n = 49 (11.0%) |
| Chronic pain | - | n = 77 (17.3%) |
| Other impairments | - | n = 25 (5.6%) |
| **Paid work** | Missing n = 0 | Missing n = 95 |
| Yes | n = 101 (28.6%) | n = 121 (34.6%) |
| No | n = 252 (71.4%) | n = 229 (65.4%) |
| **Social isolation raw-score** | | |
| Mean (std. error) | 24.3 (0.5) | 13.9 (0.3) |
| Std. deviation | 8.4 | 6.1 |
| Median | 24 | 12 |
| Minimum–maximum | 8–40 | 8–40 |
| 25th percentile | 18 | 8 |
| 75th percentile | 31 | 17 |

**Table 1.** *Cont.*

| Descriptive Variables | Canada (n = 353) | The Netherlands (n = 445) |
|---|---|---|
| **Social isolation T-score *** (%)** | | |
| T-scores lower than 40 | n = 17 (5%) | n = 111 (32%) |
| T-scores 40–50 | n = 44 (13%) | n = 134 (39%) |
| T-scores 50–60 | n = 147 (43%) | n = 91 (26%) |
| T-scores higher than 60 | n = 131 (39%) | n = 12 (3%) |
| **Self-reported general health (%)** | | |
| Excellent | n = 8 (2%) | n = 10 (3%) |
| Very good | n = 42 (12%) | n = 26 (7%) |
| Good | n = 125 (36%) | n = 144 (40%) |
| Fair | n = 112 (32%) | n = 146 (41%) |
| Poor | n = 59 (17%) | n = 30 (8%) |
| **Negative impact of COVID-19 on mental health (%)** | | |
| Not at all | n = 18 (5%) | n = 121 (34%) |
| Very little | n = 49 (14%) | n = 132 (37%) |
| To some extent | n = 134 (40%) | n = 65 (18%) |
| To a great extent | n = 86 (25%) | n = 23 (7%) |
| Completely | n = 51 (15%) | n = 12 (3%) |
| **Changes in lifestyle since COVID-19 (%)** | | |
| No changes in eating habits | n = 99 (30%) | n = 243 (69%) |
| Less healthy eating habits | n = 181 (54%) | n = 56 (16%) |
| Healthier eating habits | n = 54 (16%) | n = 54 (15%) |
| No changes in smoking habits | n = 15 (36%) | 18 (53%) |
| Less healthy smoking habits | n = 16 (38%) | n = 8 (24%) |
| Healthier smoking habits | n = 26 (21%) | n = 8 (24%) |
| No changes in alcohol consumption | n = 58 (46%) | n = 119 (80%) |
| Less healthy alcohol consumption | n = 41 (33%) | n = 13 (9%) |
| Healthier alcohol consumption | n = 26 (21%) | n = 16 (11%) |

Notes: * In the Canadian survey, type of disability was assessed by asking how participants would describe their disability, in which participants could select more than 1 option. ** Diagnosis/disability of Dutch participants was registered by rehabilitation professionals at baseline. *** A T-score of 50 on social isolation represents the mean score in a sample of individuals from the general population in the United States (reference population). A higher score indicates greater social isolation [41]. Supplementary File S3 includes figures illustrating the findings on general health, mental health and lifestyle changes.

### 3.2. Physical Activity

Canadian participants spent on average 100 min (SD 269.2, median 0.0) on moderate PA in the previous week, and 64 min (SD 240.6, median 0.0) on vigorous PA (see Table 2). Dutch participants spent on average 706 min (SD 867.5, median 355.0) on moderate PA per week, and 227 min (SD 328.4, median 120.00) on vigorous PA (see Table 2).

**Table 2.** Physical activity levels in Canada and the Netherlands.

| | **Canada** | | |
|---|---|---|---|
| | Moderate Physical Activities in Minutes in Past Week (n = 325) | Vigorous Physical Activities in Minutes in Past Week (n = 330) | Total Minutes of Moderate and Vigorous Physical Activities in Past Week (n = 330) |
| Mean (std. error) | 99.9 (14.9) | 64.2 (13.2) | 162.6 (23.0) |
| Std. deviation | 269.2 | 240.56 | 417.5 |
| Median | 0.0 | 0.0 | 0.0 |
| Minimum | 0.0 | 0.0 | 0.0 |
| Maximum | 3360.0 | 2880.0 | 3540.0 |
| 25th percentile | 0.0 | 0.0 | 0.0 |
| 75th percentile | 90.0 | 15.0 | 120.0 |

**Table 2.** *Cont.*

| | The Netherlands | | |
|---|---|---|---|
| | **Moderate physical activities in minutes per week (n = 409)** | **Vigorous physical activities in minutes per week (n = 409)** | **Total minutes of moderate and vigorous physical activities per week (n = 409)** |
| Mean (std. error) | 706.2 (42.9) | 227.3 (16.2) | 933.5 (49.3) |
| Std. deviation | 867.5 | 328.4 | 996.6 |
| Median | 355.0 | 120.0 | 600.0 |
| Minimum | 0.0 | 0.0 | 0.0 |
| Maximum | 4920.0 | 2580.0 | 4980.0 |
| 25th percentile | 60.0 | 0.0 | 180.0 |
| 75th percentile | 1080.0 | 325.0 | 1305.0 |

Notes: In Canadian survey, self-reported physical activity was measured using IPAQ-SF [36]. In the Dutch survey self-reported physical activity was measured using the Adapted-SQUASH [37].

Of the Canadian participants that participated in PA (N = 242), 64.5% reported that they became less physically active since the start of the pandemic, 13.2% reported to have become more physically active, while 22.3% reported no change in their PA.

Of the Dutch participants that participated in PA (N = 335), 37.0% reported that they became less physically active during leisure time, while 18.5% became more active, and 44.5% reported no change in leisure time PA. Supplementary File S4 provides additional details on changes in PA since the start of the pandemic.

*3.3. Social Isolation*

Table 1 shows the social isolation score (raw scores and T-scores) among Canadian and Dutch participants. 82.0% of Canadian participants experienced greater social isolation than the reference population in the US, compared to 29.0% of the Dutch participants.

*3.4. General Health, Mental Health and Lifestyle*

17.1% of Canadian participants reported having poor general health, compared to 8.4% of Dutch participants. 2.3% Canadians indicated having excellent general health, compared to 2.8% of Dutch participants.

94.7% of Canadian participants reported experiencing some negative impact of COVID-19 on their mental health (ranging from very little to completely impacted), compared to 65.7% of Dutch participants.

As for change in lifestyle, the eating habits of more than half of Canadian participants (54.2%) became less healthy, compared to 15.9% of Dutch participants. About the same amount of Canadian and Dutch participants started eating healthier (16.2% and 15.3%, respectively). By far the largest part of Dutch participants experienced no change in eating habits (68.8%), compared to 29.6% of Canadians. 38.1% of Canadian participants that smoked, started smoking more since the start of the pandemic, compared to 23.5% of Dutch participants that smoked. 32.8% of Canadian participants started drinking more alcohol, compared to 8.8% of Dutch participants. In both countries, many participants experienced no change in alcohol consumption (Canada: 46.4%; Netherlands: 80.4%).

*3.5. Associations between Physical Activity and Social Isolation*

Table 3 reports on the crude and adjusted associations between PA and social isolation for the Canadian and Dutch samples separately. Results indicate no clinically relevant associations between PA and social isolation. To illustrate, in the Canadian sample, a 1 h increase in PA was associated with a non-significant decrease of −0.090 (−0.257–0.050) points in social isolation.

**Table 3.** Associations between physical activity and social isolation for the Dutch and Canadian samples separately.

| | Netherlands | | Canada | |
|---|---|---|---|---|
| | B, 95% Confidence Interval | *p*-Value | B, 95% Confidence Interval | *p*-Value |
| Physical activity [#], crude | −0.020 (−0.059–0.19) | 0.310 | −0.068 (−0.245–0.069) | 0.328 |
| Physical activity, adjusted * | 0.001 (−0.040–0.042) | 0.975 | −0.090 (−0.257–0.050) | 0.199 |

Notes: [#] Physical activity is presented per hour. * Adjusted for age, gender and marital status.

## 4. Discussion

This study provides new insights into the negative impact of the COVID-19 pandemic on PA, social isolation, general health, mental health and lifestyle among adults with physical disabilities living in Canada and the Netherlands. Overall, Dutch participants reported higher levels of PA, lower levels of social isolation, and better general health during the second year of the pandemic, compared to Canadian participants. In both samples, no clinically relevant associations were found between PA and social isolation.

### 4.1. Physical Activity

The percentage of the participants that reported to be less physically active since the start of the pandemic was high among both Canadian participants and Dutch participants (64% vs. 37%, respectively). This negative impact of the pandemic on PA levels was reported in other studies among people with and without disabilities across the world [6,8]. We found large differences in PA levels between Canadian and Dutch participants (163 min vs. 934 min of PA). PA levels of the Canadian participants were generally low, but comparable to findings from other studies on measured self-reported PA among people with physical disabilities [9,10,13,45]. In contrast, PA levels of Dutch participants were high compared to other Dutch studies on self-reported PA among people with physical disabilities and/or chronic diseases [46–48]. The Dutch Adapted-SQUASH may overestimate participants' PA levels more than the IPAQ-SF because the Adapted-SQUASH includes items on various PA settings. The differences between countries may therefore be explained by differences in questionnaires. To consider these differences in questionnaires, we specifically focused the analyses on the most robust PA-measures: moderate and vigorous PA. Dutch participants were still consistently more active compared to the Canadian population, but comparisons should be made with caution.

A possible explanation for these high PA levels and relatively lower negative impact of the pandemic reported by Dutch participants is that the Dutch survey was conducted as part of the ongoing ReSpAct study. ReSpAct participants took part in the RSE program in which they received PA counseling during and after their rehabilitation treatment 6–8 years ago [30]. They were supported to maintain a physically active lifestyle after discharge from rehabilitation [30]. While the Dutch cohort may be a selective, non-representative sample in terms of their PA interests and motivation towards an active lifestyle [30], it is encouraging that many of the Dutch participants reported high levels of PA during the pandemic. Our findings from the 6–8 years follow-up study are promising in terms of the potential long-term outcomes of integrating PA promising strategies during and after rehabilitation. Indeed, the Netherlands has a history of programs focusing on integrating and promoting PA in rehabilitation and hospital settings. These findings and previous studies may inspire other groups in other countries to integrate PA promotion strategies in rehabilitation and hospital settings in order to improve PA levels in PWD [29,49,50].

The Canadian participants were recruited as part of the COVID-19 Disability Study, a unique research partnership initiative to record the experiences, needs and concerns of PWD living in Canada during the pandemic. While this study did not have a specific focus on PA, the findings of the COVID-19 Disability Study have been used to inform policy

changes in Ontario regarding access to PA programs specific for PWD. Recreation facilities could re-open for PWD who needed physical therapy. This is an example of how research findings could inform policy changes and immediately have a positive influence on lives of many Canadians with disabilities. This impactful initiative may be explained by the fact that the COVID-19 Disability Survey has been designed, conducted and disseminated in partnership between researchers and relevant research users (i.e., community organizations). Furthermore, the findings were shared via accessible, open-access reports available on www.disabilitysurvey.ca (accessed on 7 December 2022). Together, the COVID-19 Disability Survey provides an example of how a research partnership approach may contribute to improving the translation of research findings to practice and policy, and may inspire other groups to apply a similar research partnership approach to their (future) projects [51].

### 4.2. Social Isolation

Canadian participants reported higher levels of social isolation than Dutch participants (82% vs. 29%) during the second year of the pandemic. In a study conducted in the UK, 49% of the PWD reported to feel lonely during the pandemic [24]. An increase in feelings of loneliness and social isolation during the pandemic has been reported in previous studies among older adults and PWD [10,45,52]. The generally high levels of social isolation among Canadian participants in comparison with the Dutch participants may be explained by differences in implemented COVID-19 and lockdown measures. The data collection for the Canadian study started in December 2020 when COVID-19 restrictions were still in place in many provinces. The data collection for the Dutch study took place between March–August 2021, a period where many COVID-19 restrictions were lifted. As such, it is possible that Dutch participants were less negatively impacted by the pandemic compared to Canadians due to the differences in policies and lockdown measures. Another explanation for the difference in social isolation may be the higher percentage of Dutch participants (69.6%) reporting to be married compared to the Canadian participants (34.6%). Marital status has been linked to social isolation [53]. Similarly, Dutch participants reported higher levels of PA, which may contribute to lower levels of social isolation, as supported by various theoretical frameworks (e.g., Broaden-and-Build Theory of Positive Emotions, Social Support Theory). However, this explanation is not supported by our findings on the associations between PA and social isolation.

The lack of clinically relevant associations between PA and social isolation was in contrast to our hypothesis. This may be due to the multidimensional aspect of both PA and social isolation. For example, we only looked at PA on moderate and vigorous intensity, while PA on mild intensity (e.g., walking/wheeling) may also be associated with social isolation. However, findings of previous studies examining relationships between PA and social isolation were also mixed [21,22]. Future (longitudinal) research is needed to understand if and how PA can be used as an effective strategy to reduce social isolation and other mental health distress in PWD, both during and after the pandemic.

### 4.3. General Health, Mental Health and Lifestyle

More than half of Canadian participants (53%) and 44% of the Dutch participants reported their current health as fair or poor. Previous studies conducted before and during the COVID-19 pandemic showed that PWD reported lower levels of health and well-being compared to those without disabilities [26,45,52]. Various studies from across the world reported the negative impact of the pandemic on health and well-being in PWD [7–9,45,52]. Indeed, our results showed that almost all Canadian participants (95%) reported to some negative impact of COVID-19 on their mental health, compared to 72% of Dutch participants. The higher percentages of the Canadians reporting a negative impact on their mental health compared to the Dutch participants may be related to the differences in PA levels. A systematic rapid review on the association of PA with depression and anxiety found that engaging in PA during the pandemic was associated with lower levels of depression and anxiety [14]. Differences in COVID-19 measures between countries could

also explain differences in perceived (negative) impact between countries. As countries differ in pandemic response strategies and infection waves do not happen simultaneously across countries [54], differences in reported negative impact on mental health could be partly attributed to that. Future research is needed to longitudinally monitor mental health of PWD during, but also after the pandemic and its relationships with PA, and other lifestyle behaviours.

Regarding changes in other lifestyles, we found that Canadians reported less healthy eating and more alcohol and tobacco consumption since the pandemic more often compared to the Dutch participants. A systematic review among people aged 16 and older on snacking behaviour, fast-food consumption, and alcohol consumption during the pandemic showed that increased snacking was found for a significant portion of the population (18.9–45.1%), whereas fast-food consumption showed a tendency towards decrease (15.0–41.3%) [55]. In 17 out of 23 studies, alcohol consumption did not change during the lockdown for most participants [55]. Grossman, Benjamin-Neelon and Sonnenschein [56] reported that 60% of US adults over 21 years reported increased drinking compared to before the pandemic. Reasons for increased drinking included increased stress (45.7%), increased alcohol availability (34.4%), and boredom (30.1%). Although the prevalence of increased alcohol consumption during the pandemic was lower in our samples, the percentage of participants reporting to have increased their alcohol consumption during the pandemic is still worrisome. They do partly overlap with findings from the UK [45] and emphasize the need to develop multiple lifestyle programs and interventions to support PWD, but also people without disabilities in starting and maintaining a healthy lifestyle during and after the pandemic.

### 4.4. Limitations and Future Directions

Some limitations need be acknowledged. First, both studies used different criteria to include participants with physical disabilities and/or chronic diseases. As such, study populations in the Canadian and Dutch study were not fully identical due to these differences in recruitment and promotion strategies between countries. We were also unable to do any sub-analyses focusing of specific groups of disabilities as type of disability/diagnosis was measured in different ways. Another limitation is that we did not collect detailed information on different mental health components, such as anxiety, depression, or stress. Furthermore, we only reported on changes in PA, eating, smoking and alcohol consumption, and did not collect data on other lifestyle factors, such sleep or stress management activities. Moreover, PA was not assessed by the same questionnaire in both samples, making direct and valid cross-country comparisons of PA difficult or even impossible. Another limitation of this study is the cross-sectional design of the study. We examined the associations between PA and social isolation cross-sectionally. As such, we cannot determine any cause-and-effect relationships between PA and social isolation related to the pandemic. Finally, this study provides a snapshot of the impact of the pandemic on PA, social isolation, general health, mental health and lifestyle among people with physical disabilities in Canada and the Netherlands. The pandemic is, unfortunately, not over. As reduced PA could lead to progression of disablement in older or diseased populations, further research on the aftermath of the pandemic for people with physical disabilities is recommended. It is important to identify what long-term or sustained consequences the pandemic has on the lifestyle, general- or mental health and social isolation of people with physical disabilities, as well as the consequences of the decrease in PA for this group on the longer term.

### 4.5. Implications

This multi-country study is the first that systematically identified and compared the impact of the pandemic on PA, social isolation, general health, mental health and lifestyle changes in a vulnerable population of adults with physical disabilities living in Canada and The Netherlands. Although many studies have been conducted on the impact of the

COVID-19 pandemic, still a limited amount studied the impact of the pandemic on diverse group of adults with physical disabilities. This study adds to the existing literature by understanding how the pandemic impacted a diverse, active and less active group of adults with physical disabilities in different settings (Canada and the Netherlands), with different PA promotion backgrounds. We also add to the existing literature by examining associations between PA and social isolation in these two unique groups of PWD. As described in the previous section, both the Canadian and Dutch studies are unique, nationwide initiatives that may inspire other groups from across the world to initiative activities to improve PA levels in PWD (e.g., PA promotion in rehabilitation/hospital, conducting and disseminating research in partnership, publishing open-access, accessible reports). In addition to the new insights into differences between countries, cross-country comparative studies provide opportunities to inspire and learn from initiatives, and programs implemented in other countries. Cross-country comparative studies can promote and improve international collaborations between research groups.

A practical example of how international collaborations can enhance the impact of research findings and improve support to PWD is the implementation of the Canadian PA Coaching Service, called Get in Motion, in the Netherlands. Get in Motion is an evidence-based phone-based PA coaching service that was developed in 2008 to support people with spinal cord injury to engage in a PA [57]. At the start of the COVID-19 pandemic, the Get in Motion PA coaching service was re-launched to support Canadians with physical disabilities during the pandemic [58]. The service offers unique opportunities for people with physical disabilities to connect with volunteer coaches, and may also have positive impact on clients' general health and social isolation via these social connections. In the second year of the pandemic, the Dutch Ministry of Health, Welfare and Sports provided financial support to transfer the Get in Motion program to the Netherlands. Indeed, the service has now been adapted to the Dutch context and will be ready for official launch in the fall of 2022. In sum, the Dutch implementation of the Canadian Get in Motion service is a clear example of how international collaborations has the potential to improve research impact and enhance many lives of PWD. We encourage other groups to start and continue similar initiatives to enhance collaborations between countries across the world, including collaborations between low-, middle- and high-income countries.

## 5. Conclusions

This study provides a first insight into the negative impact of the COVID-19 pandemic on PA, social isolation, general health, mental health, and lifestyle changes among a diverse group of adults with physical disabilities living in Canada and the Netherlands. In both studies, a large part of participants reported a decrease in their PA since the start of the pandemic. In line with our hypotheses, our findings suggest that Canadian participants reported a greater negative impact than Dutch participants. These findings should be interpreted with caution, as the Dutch ReSpAct cohort is known as an active and motivated cohort and the Adapted SQUASH and the IPAQ PA questionnaires cannot be directly compared. Future research is needed to better understand if and how PA can be used to reduce social isolation in PWD. This study illustrates how cross-country collaborations and exchange provide opportunities to inspire and learn from initiatives and programs in other countries and may help to improve PA support among people with disabilities during and after the pandemic.

**Supplementary Materials:** The following supporting information can be downloaded at: https://www.mdpi.com/article/10.3390/disabilities2040054/s1, File S1: Details on recruitment methods of ReSpAct 2.0 study; File S2: Selection of confounders; File S3: Figures of results, and File S4: Change in physical activity. References [59–65] are cited in the supplementary materials.

**Author Contributions:** Conceptualization, T.H., F.H., COVID-19 Disability Survey Group and the ReSpAct 2.0 Group; methodology, K.M., T.H. and F.H.; software, K.M., T.H. and F.H.; validation, all authors; formal analysis, K.M., T.H. and F.H.; investigation, K.M., P.B. and F.H.; resources, COVID-

19 Disability Survey Group and the ReSpAct 2.0 Group; data curation, K.M. and P.B., COVID-19 Disability Survey Group and the ReSpAct 2.0 Group; writing—original draft preparation, K.M.; writing—review and editing, all authors; visualization, K.M. and F.H.; supervision, T.H. and F.H.; project administration, COVID-19 Disability Survey Group and the ReSpAct 2.0 Group; funding acquisition, COVID-19 Disability Survey Group and the ReSpAct 2.0 Group. All authors have read and agreed to the published version of the manuscript.

**Funding:** The ReSpAct study was funded by the Dutch Ministry of Health, Welfare and Sports (grant no. 319758), Stichting Beatrixoord Noord-Nederland (ReSpAct 2.0; grant date 19 February 2018) and supported by the Knowledge Center of Sport Netherlands and Stichting Special Heroes Nederland (before January 2016: Stichting Onbeperkt Sportief). F.H. is supported by the Craig H. Neilsen Foundation Postdoctoral Fellowship (#719049) and Michael Smith Foundation for Health Research (MSFHR) Trainee Award (#RT-2020-0489).

**Institutional Review Board Statement:** The study was conducted in accordance with the Declaration of Helsinki. The Research Ethics Board of the University of British Columbia gave approval for the COVID-19 Disability Survey (Project code: H20-01203, date: 21 April 2020). The Central Ethics Committee of the University Medical Center Groningen (UMCG) gave approval for the ReSpAct 2.0 study (Project code: 201900645, date: 9 March 2020).

**Informed Consent Statement:** Informed consent was obtained from all participants involved in the study.

**Data Availability Statement:** The data that support the findings of this study are available on request from the corresponding author.

**Acknowledgments:** The COVID-19 Disability Survey Group includes: Pinder DaSilva, Femke Hoekstra, Cameron M. Gee, Tara Joy Knibbe, Emilie Michalovic, Meagan O'Neill, Adrienne R. Sinden, Joan Úbeda-Colomer and Kathleen A. Martin Ginis. The ReSpAct 2.0 Group includes: Pim Brandenbarg, Rienk Dekker, Florentina Hettinga, Trynke Hoekstra, Femke Hoekstra, Leonie Krops, Bregje Seves, and Lucas van der Woude.

**Conflicts of Interest:** The authors declare no conflict of interest. The funders had no role in the design of the study; in the collection, analyses, or interpretation of data; in the writing of the manuscript; or in the decision to publish the results.

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
