# Peer review of "The Impact of the COVID-19 Pandemic on Physical Activity and Social Isolation among Adults with Physical Disabilities Living in Canada and The Netherlands"

_disabilities, doi:10.3390/disabilities2040054_

Round 1

Reviewer 1 Report (Previous Reviewer 3)

I thank the authors for their efforts in addressing the reviewers’ comments. The suggested additional analysis from the authors regarding chronic pain is not needed as I agree that it is outside the scope of the paper. The authors also have identified this as a limitation, which is satisfactory. With the change in research focus, I believe that this paper is ready to be published.

Author Response

Thank you reviewing our manuscript and the positive comments. 

Reviewer 2 Report (New Reviewer)

I am statistician who, for several years, have supported researchers with similar questions about people with disabilities. My primary concern is the large discrepancy between of physical activity of Canadians and Dutch participants. Actually, you report other differences which you explain away by the dates of the studies, current health, and lifestyle changes. These limitations, as pointed out in lines 127-129, do not allow for valid comparisons of the results from the two countries. I suggest an effort to determine and compare pre and post covid change scores of the dependent variables of interest.

Author Response

Please find our responses to reviewer's comments in the attached file. 

Round 2

Reviewer 2 Report (New Reviewer)

I was the statistical reviewer of your previous submission. You have adequately met my concerns.

This manuscript is a resubmission of an earlier submission. The following is a list of the peer review reports and author responses from that submission.

Round 1

Reviewer 1 Report

Comments:

This cross-sectional study assesses the perceived impact of the COVID-19 pandemic on physical activity, health, and social isolation for adults with physical disabilities living in Canada and the Netherlands. A sample of 798 participants from two studies was included in this secondary data analysis. Significant between-group differences were found for social isolation. The authors conclude that Canadian participants reported a greater negative impact of the pandemic on their mental health, lower levels of physical activity, poorer general health, and greater social isolation compared to Dutch participants.

Major concerns. There are some significant limitations of this manuscript including a lack of scientific premise or hypothesis.  It is not clear why a comparison of two different samples in two different countries will assist in future policy planning.  While it is interesting to understand how the pandemic impacted people with disabilities, it is not clear how this manuscript provides clarity in how to address the negative consequences of the pandemic for people with disabilities. Additional issues related to methods and results (eg comparing two very different samples of people with disabilities on similar outcomes, lack of clarity in presenting results, etc) further dampen enthusiasm

Specific areas for improvement include:

Title:

·         The use of “perceived” is confusing – is this referring to perceptions of participants in this study?

Abstract:

·         The definition of “health” is unclear – only mental health is mentioned here, but the body of the manuscript also references general health, lifestyle, and well-being. Please be consistent with use of this term.

·         Line 29 – the use of “became” implies objective data rather than self-reported changes

Introduction:

·         The introduction lacks a scientific premise. There are inconsistencies in terms related to physical activity (eg  reference to disability sports). Physical activity and engaging in sports are different constructs.

o   Page 2, line 61 – the relevance of stress, mood, and sleeping patterns is unclear as those outcomes are not included in the presented data.

Materials and Methods:

·         (see abstract) A clear definition for “health” used consistently throughout would provide clarity to readers.

·         There is no scientific justification for the 3 factors (eating, smoking, and alcohol consumption) included in the lifestyle composite? This seems to exclude those activities that positively influence to one’s lifestyle, such as hobbies or restorative activities.

·         There is no description of how missing data were handled.

Results:

·         There is no justification for the age categories presented in Table 1.

·         The use of decimals and labeling of figure axes is inconsistent.

·         Tables and figures lack notes.

·         Table 3: There is no rationale for choosing multinomial logistic regression rather than ordinal regression, given  the response categories are ordered.

o   It is not clear why there are two identical tables with different data reported.  

o   There was not adjustment for multiple comparisons

·         Figures 1, 2, & 3 are difficult to interpret; and might be more understandable as a table.

·         The use of raw scores instead of T-scores in analysis of PROMIS social isolation is not recommended, as this can impact precision and interpretability of scores (see Hanmer, J., Jensen, R.E., Rothrock, N. et al. A reporting checklist for HealthMeasures’ patient-reported outcomes: ASCQ-Me, Neuro-QoL, NIH Toolbox, and PROMIS. J Patient Rep Outcomes 4, 21 (2020). https://doi.org/10.1186/s41687-020-0176-4)

o   The presentation of the T-score data feels misplaced; consider combining with previous group comparison data in Table 1.

Discussion:

·         Information presented in the limitations section is redundant and presented twice.  

Conclusions:

·         A more cautious interpretation of the results is warranted.

o   Page 15, line 161 – The use of “general support” is not supported by the data presented; the type of support should be specified

General:

Copy editing by a native English speaker (for clarity and grammar) would strengthen the manuscript.

Reviewer 2 Report

The research follows the basic principles of good research and is based on solid concepts. It could be improved by including a brief theoretical framework on the themes under study, as well as substantiating the conclusions, referring for example to ways to support and promote physical activity among people with physical disabilities. The way in which they explain the reason for choosing the countries under study is remarkable.

Reviewer 3 Report

This is a well-written manuscript and interesting study. Before publication, the below specific comments should be addressed.

Specific comments:

Abstract – A sentence or two should be added to the results/conclusion to state why the differences were found. As it stands, the results do not logically lead to this concluding sentence: “The findings emphasize the importance of supporting and promoting PA among people with physical disabilities during and after the pandemic”.

Abstract – Add a sentence to explain why the results should be “interpreted with caution”.

Introduction – Please add an example or two of the barriers mentioned in this sentence “The low levels of PA in PWD can be explained by the many barriers they experience to engage in PA”.

Introduction – This phrase needs a reference(s): “as both countries are high-income countries that play leading roles in expanding our knowledge on rehabilitation and disabilities sports”.

Materials and Methods, 2.1 and 2.2. – The Dutch study specifies disabilities for inclusion while the Canadian study is more general, which makes it difficult to compare. Is there any way of organising this data so that disabilities are comparable?

Materials and Methods, 2.3.1/ Data Analysis, 2.4.1  – These sections mention that questions for perceived impact on PA was different for Canada and the Netherlands: Canada had a single general question while the Netherlands had 4 questions. To have a more accurate comparison, it would be better to get an average result for the 4 Dutch questions to compare against the single Canadian question for each available answer. The analysis of the 4 questions can then be presented as a secondary analysis.

The PA results are concerning due to the considerable differences between the Canadian results and the Dutch results. The median results in particular are red flags to me as the Canadian medians are all 0 while the Dutch medians are in triple digits. As the authors have stated, the Dutch study explicitly encouraged PA while the Canadian study did not; I find this to be a major flaw that does not make the PA results that useful. Generally, those who engage in more PA tend to have a healthier lifestyle and better mental health. This means that the other results within this study are skewed due to this flaw. Is there any way to rectify this, such as only looking at data from the beginning of the Dutch study (i.e. prior to the participants engaging in PA counselling)?

Another factor that may have impacted the findings is the marriage status of the participants, where the Dutch participants were more likely to be married/living as if married than the Canadian participants. Assuming that the Dutch participants and their partners actually like each other, the social support may have given them more opportunities to engage in PA and a healthier lifestyle. Having more social support also tends to improve mental health.

Beyond the collected data and the differences in COVID-19 restrictions, I request the authors to briefly discuss other differences that may have contributed to the results. For example, are the social systems and health systems similar between Canada and the Netherlands? I realise that this is outside the aim of the current study, but I believe a reference to factors that affect PA and healthy lifestyle accessibility are important to consider, especially for future research.
